# Possibilities of Decreasing Hygroscopicity of Resonance Wood Used in Piano Soundboards Using Thermal Treatment

Petr Zatloukal [1] , Pavlína Suchomelová [1], Jakub Dömény [1], Tadeáš Doskočil [2], Ginevra Manzo [1] and Jan Tippner [1,*]

[1] Faculty of Forestry and Wood Technology, Mendel University in Brno, Zemědělská 3, 613 00 Brno, Czech Republic; xzatlou6@node.mendelu.cz (P.Z.); pavlina.suchomelova@mendelu.cz (P.S.); jakub.domeny@mendelu.cz (J.D.); ginevra.manzo@mendelu.cz (G.M.)

[2] Petrof, spol. sr.o, Na Brně 1955, Nový Hradec Králové, 500 06 Hradec Králové, Czech Republic; doskocil@petrof.com

* Correspondence: jan.tippner@mendelu.cz

**Abstract:** This article presents the possibilities of decreasing moisture sorption properties via thermal modification of Norway spruce wood in musical instruments. The 202 resonance wood specimens that were used to produce piano soundboards have been conditioned and divided into three density groups. The first specimen group had natural untreated properties, the second was thermally treated at 180 °C, and the third group was treated at 200 °C. All specimens were isothermally conditioned at 20 °C with relative humidity values of 40, 60, and 80%. The equilibrium moisture content (*EMC*), swelling, and acoustical properties, such as the longitudinal dynamic modulus ($E'_L$), bending dynamic modulus ($E_b$), damping coefficient (*tan δ*), acoustic conversion efficiency ($ACE_L$), and relative acoustic conversion efficiency ($RACE_L$) were evaluated on every moisture content level. Treatment at 180 °C caused the EMC to decrease by 36% and the volume swelling to decrease by 9.9%. Treatment at 200 °C decreased the EMC by 42% and the swelling by 39.6%. The 180 °C treatment decreased the value of the longitudinal sound velocity by 1.6%, whereas the treatment at 200 °C increased the velocity by 2.1%. The acoustical properties $E_L'$, $E_b$, $ACE_L$, and $RACE_L$ were lower due to the higher moisture content of the samples, and only the *tan δ* increased. Although both treatments significantly affected the swelling and *EMC*, the treatment at 180 °C did not significantly affect the acoustical properties.

**Keywords:** acoustic properties; hygroscopicity; sound velocity; spruce; swelling; thermal treatment

## 1. Introduction

Wood is commonly used to make musical instruments across the world, and until today, there has been no available substitute for this superb material. The natural hygroscopicity and related hygroexpansion are the main disadvantages of using wood in musical instruments.

Resonant spruce (high-quality *Picea abies* L. Karst) is a popular material for piano soundboard fabrication [1]. Resonance spruce wood is generally characterized by narrow annual rings (at least four in one centimeter), and it usually grows on either poor soil in high-altitude plateaus or on north-oriented hillsides. High-quality resonance wood for soundboards is characterized by high sound velocity in a longitudinal direction, low internal friction, relatively low density, high radiation ratio, and a dynamic modulus or specific Young's modulus $E_L/δ$.

It is generally known that air humidity causes pianos to become untuned, as swelling causes the soundboard to move, which affects the strings [2]. Therefore, the frequencies change over time. The dimensional instability of wood under different humidity conditions can increase the number of services due to untuning or defects in a piano. There are several possibilities to stabilize the conditions inside the instruments.

For example, [3] describes a humidifier/dehumidifier device that can regulate humidity with an accuracy of 1%. Piano users state that these solutions are insufficient, especially in conditions with significant humidity changes.

There are several specialized wood modifications that can be applied in order to improve certain wood properties [4].

Active and passive way of wood modification can be applied. In general, the passive one-lumen filling modification highly influences the physical and mechanical properties. The less invasive cell wall active modification involves reactions with polymers, cross linking, and degradation of the cell wall. From the group of chemical modifications are known cross-linking, bulking, grafting, and degradation of cell wall [5].

Furfurylation is the impregnation of wood with furfuryl-alcohol whilst increasing the temperature during the reaction [5]. The raised temperature then acts as a reaction catalyst. Although this process prevents humidity absorption, it increases density, which decreases the acoustic conversion efficiency ($ACE_L$) and relative acoustic conversion efficiency ($RACE_L$) [6,7]. Additionally, the DiMethylolDihydroxyEthyleneUrea (DMDHEU) resin and thermo-hydro-mechanical (THM) treatment decrease the hygroscopicity; however, this leads to an undesirable increase of the wood density [4,8]. Acetylation is an anhydride vapor curing process that uses catalysts, such as pyridine or sodium acetate. It decreases both swelling and sound velocity. The $ACE_L$ and quality factor $Q^{-1}$ remain almost unchanged [9–11]. Moreover, saligenin treatment or nano-solution SurfaPore$^{TM}$ improve the water-related properties and have an insignificant effect on the wood density [12,13].

These treatments are potentially suitable for the soundboard properties; however, they are technically and economically unavailable in piano production. Heat treatment is one of the most commonly used modification to improve dimensional stability and biological durability of wood [4,14]. At higher temperatures (180–260 °C), the hydroxyl groups reduced hygroscopicity and swelling [4,14]. Esteves et al. [15] modified resonance spruce at 190 °C. They found that although the acoustical properties improved, the wood became brittle. In addition, the density, Equilibrium Moisture Content (EMC), and damping coefficient decreased, and the modulus of elasticity and sound velocity increased. Wagenfuehr et al. [16] and Pfriem et al. [17] had similar results. According to Puszynski et al. [18], the most suitable temperature for modification lies between 160–180 °C. As there is a lower content of hydroxyl groups after thermal modification, the sorption is decreased [4]. Zhu et al. [19] measured acoustic-vibration parameters after heat treatment. Specific Young's modulus, the coefficient of sound-radiation resistance and the ratio of Young's modulus to the dynamic stiffness modulus increased, whereas sound resistance decreased. The best vibration performance was obtained by 4 h treatment at 210 °C. Esteves et al. [15] found that thermal modification caused a lower density (by approximately 5%) and lower EMC (about 42%). The modulus of elasticity increased by around 4%; the longitudinal sound velocity increased by around 5%. Almost identical results were published by Pfriem et al. [17] and Pfriem [20]. This kind of modification could be used in piano production. Unfortunately, published studies related to modification techniques of wood for musical instruments are still limited. Therefore, the present study aims to find a way to decrease hygroscopicity and swelling of the piano soundboard wood, focusing on sustainability and maintaining its acoustical properties.

## 2. Materials and Methods

Resonance spruce (*Picea abies* L. Karst) specimens (44 × 12 × 450 mm) with no defects (cracks, knots, etc.) were sorted into three groups, i.e., reference group, High Temperature (HT) treated at 180 °C, and HT treated at 200 °C, with 40 specimens in each group.

At first, all specimens were conditioned at a temperature of 20 °C and an air humidity of 40% in a climatic chamber (Memmert CTC256). These conditions match with approx.

*EMC* 7.7%. Moisture content was determined using the oven-dry method according to EN 13183-1(2002) [21] and the following equation (Equation (1)):

$$EMC = \frac{1}{B} \times ln\frac{A}{ln\frac{1}{\varphi}} \tag{1}$$

where, according to de Boer–Zwicker pay: $A = 7.731706 - 0.014348*T$; $B = 0008746 + 0.000567*T$.

The hygroscopicity of the wood was evaluated via the equilibrium moisture content in both the unmodified and modified specimens using the standard gravimetric method (Equation (2)):

$$W = \frac{m_w - w_0}{m_0} \times 100 \tag{2}$$

where $m_w$—weight of saturated wood, $m_0$—weight of bone-dry wood.

The main characteristic of dimension changes is volume swelling because it includes all dimensions of the specimen. In this research paper, volume swelling was used to compare unmodified specimens with modified specimens. For this comparison, the air humidity increase from 40 to 60% was selected. Volume swelling was determined using the following equation (Equation (3)):

$$\alpha_i = \frac{a_{imax} - a_i(w)}{a_i(w)} \times 100 \ (\%) \tag{3}$$

where, $\alpha_{imax}$—dimension in any direction after swelling; $\alpha_i \ (w)$—the same dimension before swelling.

Moderate thermal modifications at 180 °C for 8 h and 200 °C for 10 h were selected for the purpose of this study. The modification course is shown in Figure 1, and the thermal modification chamber scheme is shown in Figure 2. After this modification, all specimens were climatized at a temperature of 20 °C and an air humidity of 60%. The acoustical properties and dimension changes were measured in every humidity status.

To test the dynamic properties, the sample was positioned on two flexible supports (free–free support condition). The points of support were located in the nodes (minimum amplitude) of the fundamental bending mode shape of vibration (at 0.224 and 0.776 of the sample length, which is in 100.8 and 349.2 mm length of the specimen) [22,23].

The first and second bending frequencies, longitudinal natural frequencies, logarithmic decrement of damping, and the sound velocities were measured. The flexural vibration was induced by the impact of a soft hammer to the center of the sample perpendicularly to the length (see Figure 3). The sound propagation velocity was measured using the FAKOPP Ultrasonic Timer apparatus with US10 sensors in longitudinal and transversal directions.

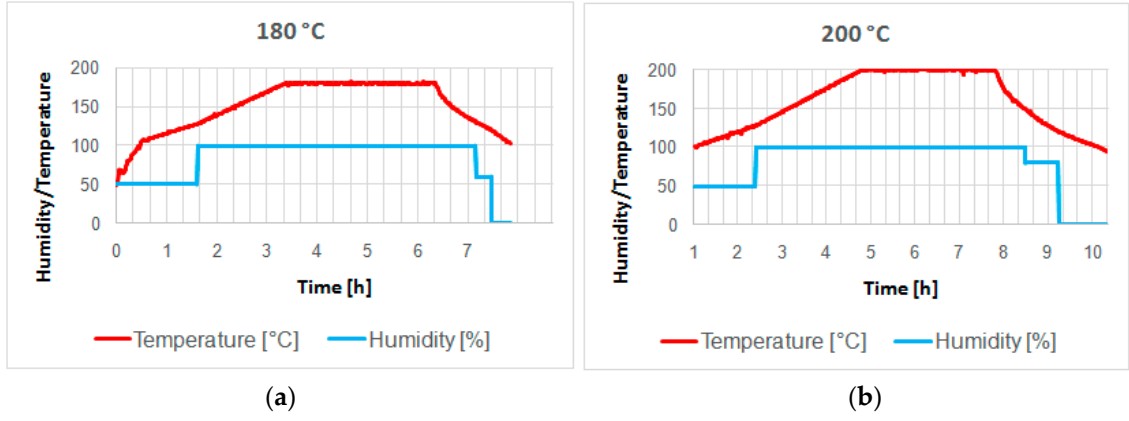

(**a**)          (**b**)

**Figure 1.** Temperature and humidity courses with the modifications (**a**) 180 °C; (**b**) 200 °C.

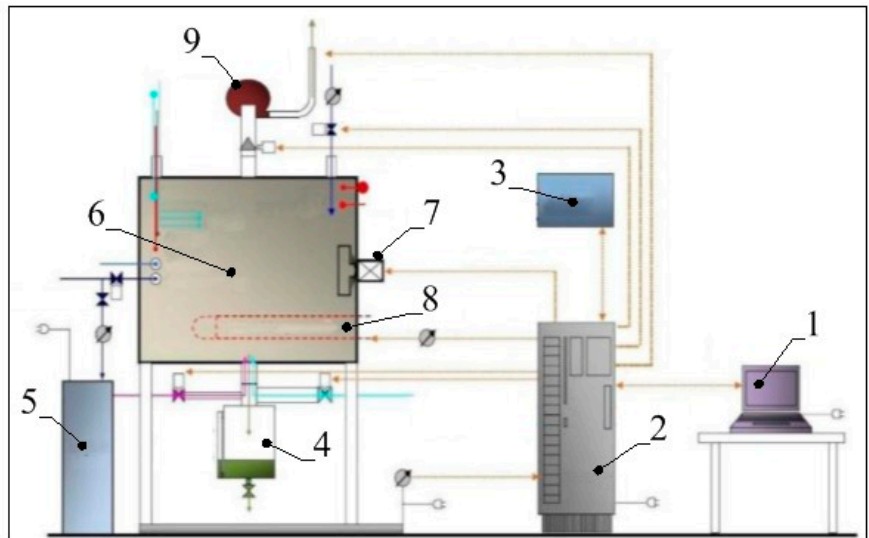

**Figure 2.** Modification chamber scheme: (**1**) PC; (**2**) switchboard; (**3**) measuring box; (**4**) condensate resorvoir; (**5**) steam generator; (**6**) thermal modification chamber; (**7**) fan; (**8**) heating element; (**9**) exhaust fan.

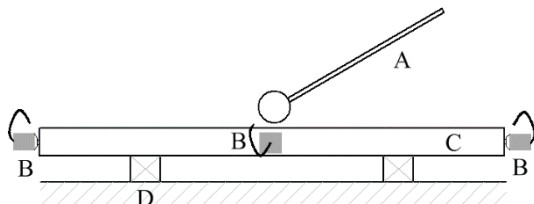

**Figure 3.** Measuring the acoustical properties: (**A**) soft hammer; (**B**) ultrasonic sensor or microphone; (**C**) specimen; (**D**) soft pads.

The derived acoustical properties are as follows:
Dynamic Young's modulus (Equation (4))

$$E_L' = c^2 \times \rho \tag{4}$$

$c$ —sound velocity; $\rho$—density.
*Dynamic bending modulus* of elasticity (Equation (5))

$$E_b = \frac{f_1^2 \times m \times l^2}{3.56^2 \times w \times \dfrac{h^3}{12}} \tag{5}$$

where $f_1$—first bend frequency; $m$—specimen weight; $l$—specimen length; $w$—specimen width; $h$—specimen thickness.
*Acoustic conversion efficiency* (Equation (6))

$$ACE_L = \frac{\sqrt{\dfrac{E_L'}{\rho^3}}}{\tan \delta} \tag{6}$$

where $E_L'$ is the dynamic Young's modulus; $\rho$—density; *tan δ*—internal friction.

A measured Logarithmic Decrement of Damping (LDD) was used to calculate the *internal friction* as follows (Equation (7)):

$$\tan \delta = \frac{LDD}{\pi} \tag{7}$$

*Relative acoustic conversion efficiency* (Equation (8)):

$$RACE_L = \frac{\sqrt{\dfrac{E_L'}{\rho}}}{\tan \delta} \tag{8}$$

Due to chemical changes, the specimens' weight reduced after the modification. The weight loss was calculated as a Mass Loss (*ML*) using the following equation (Equation (9)):

$$ML = \frac{m_1 - m_0}{m_0} \times 100 \tag{9}$$

## 3. Results and Discussion

### 3.1. Hygroscopicity

Significant differences were found among the modified and unmodified specimens. Specimens with 60% air humidity were compared, modified, and referenced during this process. The multiple comparison was determined using Scheffe test. The decrease in EMC between the groups was statistically significant. The specimens that were modified at a temperature of 180 °C showed an approximately 36% lower EMC than the unmodified specimens. The specimens that were modified at 200 °C showed an approximately 42% decrease in water content. The results are shown in Figure 4. The EMC reduction is related to the time and temperature of the process in correspondence to [4,24,25]. Akylidiz et al. [24] found reduction of EMC by 25% at 180 °C and 41% at 230 °C for black pine wood (*Pinus nigra*). Ates et al. [25] found 18% reduction of EMC at 8-h/180 °C modification and 51% reduction of EMC at 8-h/230 °C modification of calabrian pine wood (*Pinus brutia* Ten.). The EMC evaluated after modification at 180 °C are slightly higher in comparison with above mentioned results; however, higher temperature of modification brought results comparable to other researchers. The slight differences may reflect the wood species individuality (e.g., higher number of extractives in pine wood) as well as the differences in duration and temperature of treatment processes.

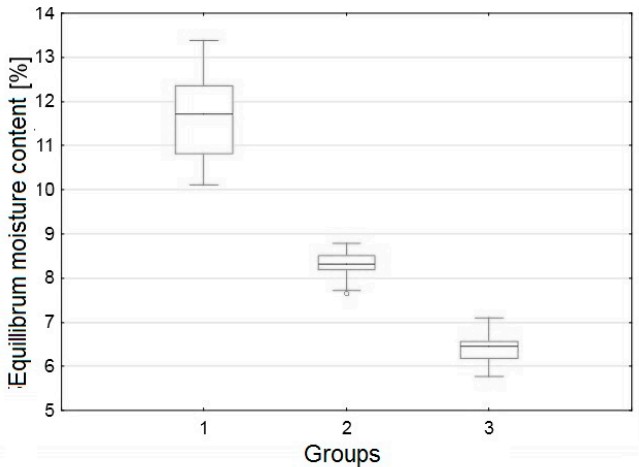

**Figure 4.** EMC of specimens: group no.1—unmodified specimens; group no.2—specimens modified at 180 °C; group no.3—specimens modified at 200 °C. EMC at 60% air humidity.

### 3.2. Swelling

In the modified specimens, the thermal modification at 180 °C decreased the volume swelling by 9.9% compared to the unmodified specimens. Moreover, thermal modification at 200 °C decreased the volume swelling by 39.6% compared to the unmodified specimens (see Figure 5). This result is crucial, as swelling is the main factor that needs to be eliminated. Swelling causes instruments to rapidly detune in changing climatic conditions [2,3]. These air humidity changes can cause various parts of the piano to crack, especially the soundboard. Scheffe test showed a statistically significant decrease in swelling in all the thermally modified groups of specimens. Swelling reduction measure, similarly to EMC, depends on used treatment parameters, e.g., temperature, time of duration, used inertial atmosphere and other standards of modification [4,15,25,26]. For example, Icel et al. [26] achieved reduction of volumetric swelling of spruce (*Picea abies*) by around 53% after modification at 212 °C and 2-h duration. Ates et al. [25] presented reduction of volumetric swelling by about 13 and 42% after 180 °C/2-h and 230 °C/8-h thermal treatment, respectively. In general, our study also confirmed that the higher time and temperature of thermal modification brings higher dimensional stability.

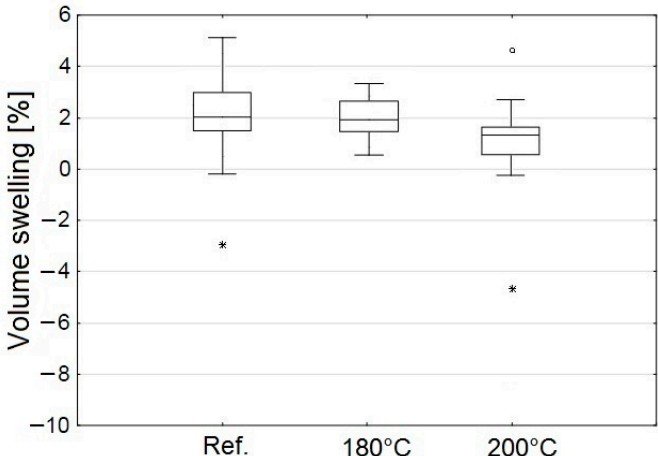

**Figure 5.** Volume swelling in the following three groups of specimens: Ref. (unmodified), modified at 180 °C, and modified at 200 °C, *—outliers.

### 3.3. Acoustical Properties

The first and second resonant frequencies and longitudinal resonant frequencies were captured using the free–free resonant method. Internal friction was also determined at this time. The rubber head stick was used for this task, and the specimen was supported by polyurethane segments using the formula $0.224 \times L$ (L = length of the specimen). These vibrations were captured using a microphone and external sound card (EDIROL FA-101) via the FireWire interface. The sound velocity was determined using the above-mentioned ultrasonic method. This measured the ultrasonic wave velocity in longitudinal and transversal directions.

Sound velocity in the longitudinal direction is affected by high moisture content. Sound velocity is significantly lower in wood with higher moisture content due to the lower Young's modulus. The higher moisture content wood contains the lower velocity sound it achieves [1,18,27].

A Scheffe test was performed. A total of seven groups were selected for this task, depending on the air humidity. These groups are described in Figure 6. The observation focused on groups 2, 4, 6 and 3, 5, 7. The only statistically significant differences were between groups 2 and 6; it means that the sound velocity was not largely affected by the modification. However, the sound velocity in the transverse direction was affected differently. In dry wood, the velocity in this direction is approximately 1200 m·s$^{-1}$. In water, it is 1485 m·s$^{-1}$ [1]. By comparing these two velocities, it is clear how increasing the

water content in the wood increases the sound velocity increases. However, in this study, we measured values with a high variance. Therefore, it is not statistically significant. The results are shown in Figure 6.

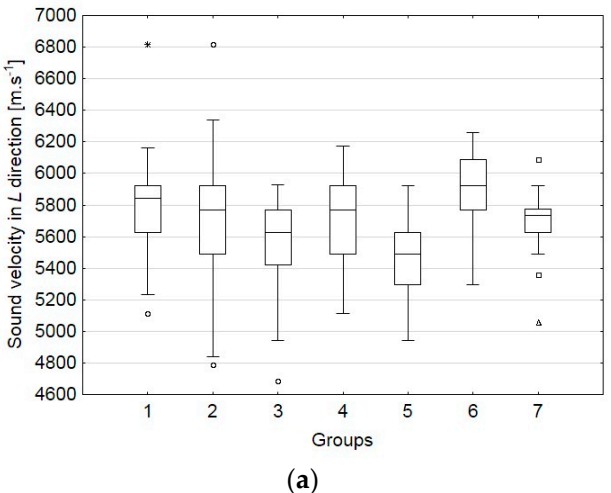
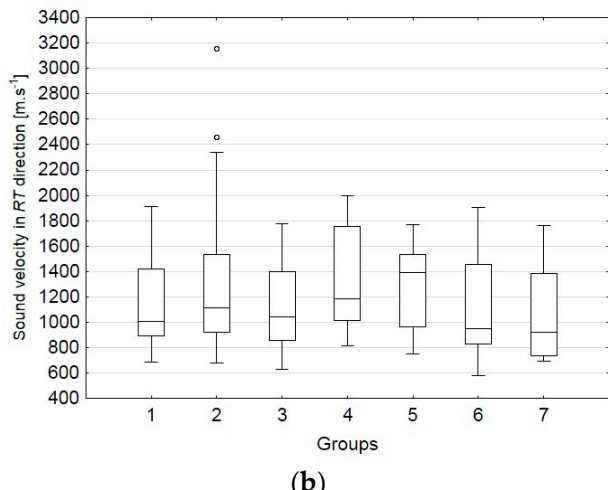

**(a)**                            **(b)**

**Figure 6.** Sound velocity in longitudinal (**a**) and transversal (**b**) directions. Group description: (**1**) 20 °C 40% unmodified; (**2**) 20 °C 60% unmodified; (**3**) 20 °C 80% unmodified; (**4**) 20 °C 60% modified at 180 °C; (**5**) 20 °C 80% modified at 180 °C; (**6**) 20 °C 60% modified at 200 °C; (**7**) 20 °C 80% modified at 200 °C.

The derived acoustical properties that were mentioned above are dependent on measured frequencies and velocities. Therefore, there is a strong correlation with these quantities. Figure 7 shows the results.

Generally, increasing the air humidity decreases the values of these properties. Otherwise, *tan δ* increases with higher moisture content and it is reduced by thermal treatment. *Tan δ* represents damping which correlates with moisture content [16,19,28,29]. Specific Young's modulus is related to sound velocity and density ratio and it was affected especially by the modification at 200 °C. The same results were reported by many other authors [16,18,19,27,29].

*ACE* and *RACE* values also depend on the moisture content in the wood. The moisture content influences density, *tan δ* and Young modulus—all input parameters for *ACE* and *RACE* definition (Equations (6) and (8)). The improvement of sonic efficiency especially with the modification at 200 °C was reported in our study. This level of thermal treatment decreases density, *tan δ* and increases Young modulus. The result is in agreement with other studies [4,19,25,27,29,30]. Treatment at temperature 180 °C did not showed significant change comparing to untreated specimens. Bending modulus of elasticity was affected by moisture content as well (the higher moisture content the lower modulus). Only modification at 200 °C causes significant increase of this parameter.

In general, based on outputs of analysis of variance the thermal modification at 180 °C did not cause any noticeable changes to the acoustical properties; however, the modification at 200 °C caused the properties to increase.

### 3.4. Weight Loss

Weight loss was detected after the thermal modification, and the referenced and modified specimens were compared. The result was an average weight loss of 4.7 and 7% after modification at 180 and 200 °C, respectively. This loss is captured in Figure 8. Alén et al. [31] found weight loss 1.5% at 180 °C and 12.5% at 225 °C treatment of spruce wood. Zaman et al. [32] determined a values of weight loss from 5.7 to 7.0% for pine wood treatment at temperature 205 °C. The weight loss is due to chemical changes in the wood. Hemicelluloses are reduced due to their thermal instability [4,33]. The higher modification temperature is used, the higher is reduction of weight [4,15,31,32].

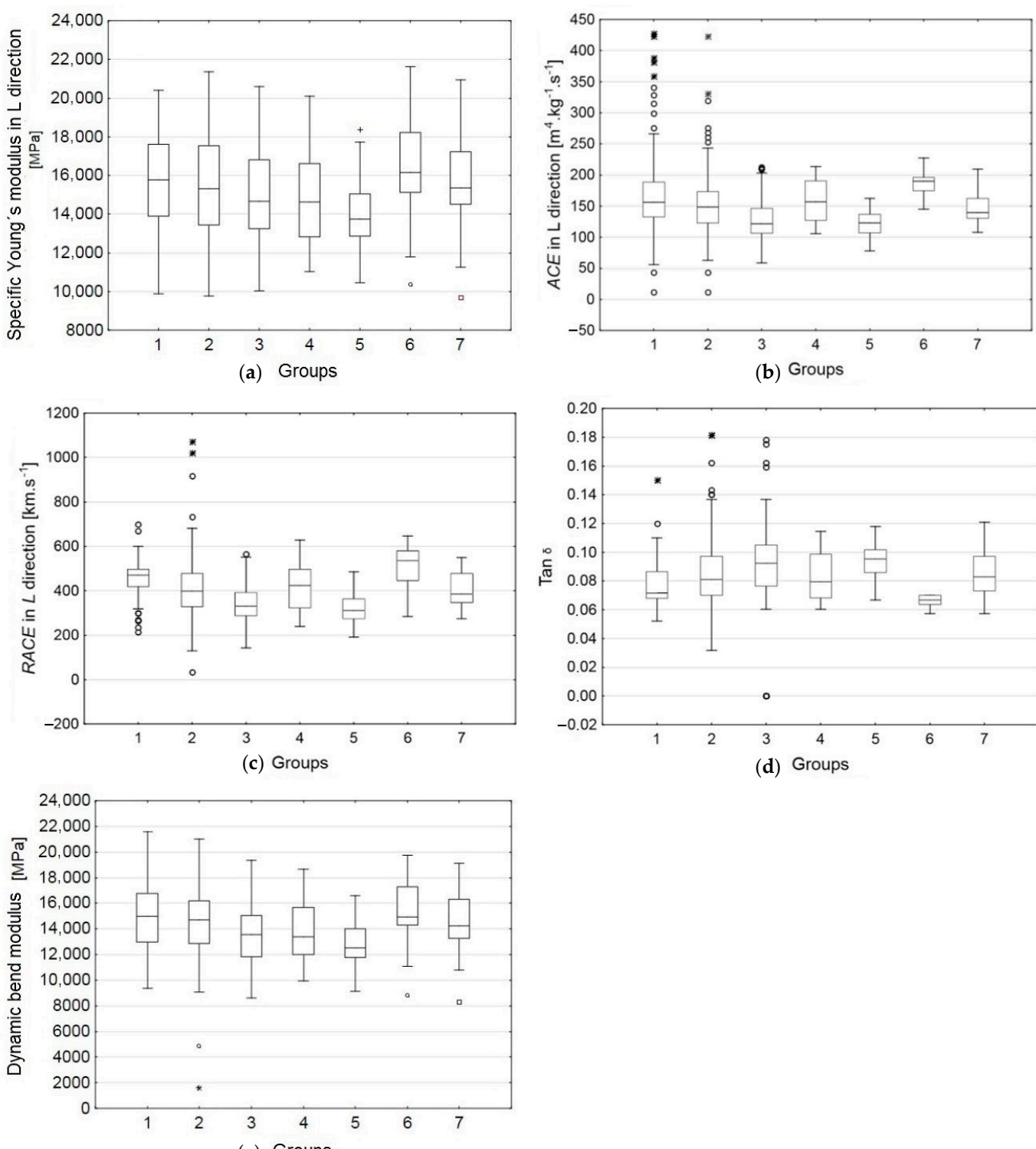

**Figure 7.** Distribution of acoustical properties in the specimens (group description is in Figure 7): (**a**) specific Young modulus; (**b**) acoustic conversion efficiency ($ACE_L$); (**c**) relative acoustic conversion efficiency($RACE_L$); (**d**) damping coefficient tan δ; (**e**) dynamic bend modulus.

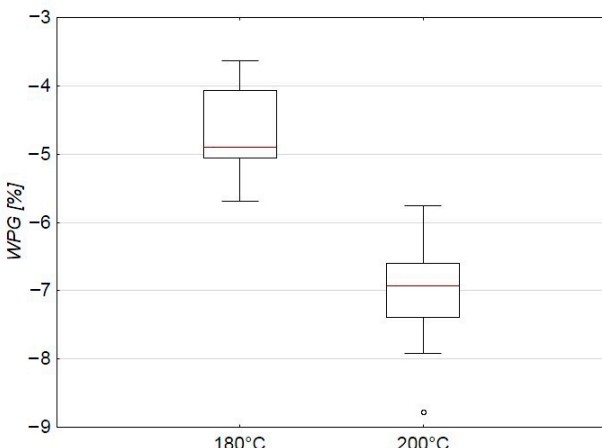

**Figure 8.** Weight loss after thermal modification in comparison to the unmodified specimens.

## 4. Conclusions

The resonance spruce specimens were thermally modified and conditioned in various conditions. The volume swelling in the modified specimens was reduced due to thermal modification at 180 and 200 °C by 9.9 and 39.6%, respectively, in comparison to the unmodified specimens.

Sorption was reduced by 36 and 42% at a modification of 180 and 200 °C, respectively. Additionally, the sound velocity in a longitudinal direction decreased with a higher moisture content.

The remaining derived acoustical properties depended on the measured frequencies, velocities, and densities. A strong correlation within these parameters was found. Generally, thermal modification at 180 °C did not cause any significant changes to the acoustical properties. The thermal modification at 200 °C had a more significant effect on the measured properties.

Thermal modification at 180 °C did not significantly affect the acoustical properties; however, the modification at 200 °C had a more significant effect. Both modifications significantly reduced swelling. The thermal modifications satisfy the requirements regarding the appearance of the musical instrument, sound quality, cost, and feasibility for the sound wood production. Two piano soundboards (for upright piano and grand piano) made of resonance spruce treated by process with parameters based on our study were produced and mounted in pianos for real-life test of sound quality and tuning stability. The darker color of soundboard brought also a positive feedback from piano designers.

**Author Contributions:** Measuring acoustical data, analyzing data, writing the manuscript P.Z.; Measuring of sorption data P.S.; Analyzing acoustic data G.M.; Performing thermal modification, process optimization J.D.; Approving the specimens´ and analyzing of acoustic data T.D.; Conceptualization, research project leader, supervising and writing the manuscript J.T. All authors have read and agreed to the published version of the manuscript.

**Funding:** This research was funded by the Technological Agency of Czech Republic project no. TH02010978 and by Petrof spol. s.r.o. company.

**Institutional Review Board Statement:** Not applicable.

**Informed Consent Statement:** Not applicable.

**Data Availability Statement:** The data are not publicly available due to privacy restrictions. The data presented in this study are available on request from the corresponding author.

**Acknowledgments:** This paper was created at the Research Center Josef Ressel in Brno-Útěchov, Mendel University in Brno with financial support from the Technological Agency of Czech Republic and Petrof spol. s.r.o company. Project "Eliminating the moisture strain on musical instrument", Reg. No. TH02010978.

**Conflicts of Interest:** The authors declare no conflict of interest.

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
