# Peer review of "Possibilities of Decreasing Hygroscopicity of Resonance Wood Used in Piano Soundboards Using Thermal Treatment"

_applsci, doi:10.3390/app11020475_

Round 1

Reviewer 1 Report

Good article, even though conclusions are not entirely clear. And the conclusions of the abstract are not fully supported by the detailed results in the paper. 

The research design is appropriate and the question is interesting. 

The results on hygroscopocity are clear and convincing. 

The results stated in the text (and in the abstract) on swelling, on the other hand are not supported by the measurement results. If Figure 5 is correct (please check!), first of all there is no statistically significant difference in swelling! The difference, at least as can be seen form the graph, does not seem significant when looking at the error bars...! 

Additionally the text is wrong, it claims a 9.9% decrease in swelling at 180°C and a 39.6% decrease of swelling at 200°C. In the graph, figure 5, the minimum is for the sample 180°C! Where are the two inverted, in figure or in text, and is minimum for 180° or 200°?

The discussion of the results on acoustical properties could be improved. Here results are claimed to be "not statistically significant" - while the differences (when comparing same moisture %) look more statistically significant than for the swelling data. 

Have some of the samples been used for piano construction? What is kind of missing is the "end of the story": which is the preferred wood batch for piano sound boards? And that would include evaluations on ease of processing as well as subjective, acoustic results. The reviewer knows that this is not evident, just a suggestion that this would of course make the article even stronger. 

Article can be published more or less at it stands - but please verify and correct Figure 5 and clearly point out whether the difference is statistically significant. 

Author Response

Dear Reviewers,

Thank you very much for your careful and thorough reading of our manuscript, we really appreciate your positive feedback and would like to thank you for constructive comments and helpful suggestions in the first round of review.

We have proofread the manuscript again and we have tried to do our best to respond to the points raised. As indicated below, we have checked all the specific comments and made the necessary changes accordingly to your indications.

Thank you again for your time.

Jan Tippner, Mendel university in Brno, 30th Decemeber 2020

Reviewer I:

• We have make clear the conflict between abstract and results: a values in the abstract regarding EMC reduction were corrected (there was incorrect values 2.8% and 4.3% for 180°C and 200°C resp.).

• Figure 5: the graph was really wrong, we are very sorry for the mistake. Values are correct (9.9% and 39.6%) but the graph was generated as incorrect plot from other measuring values. We have generated graph from the correct values again and we have reuploaded correct image.

• Discussion about acoustical properties was improved (more than 30 lines added to all sub-chapters). We have described all relevant acoustical properties and the result comparison.

• As to the last point regards to the "end of the story" we wrote final practical result of our work. Really, 2 pieces of piano soundboard from modified resonant spruce was made, now it is under long-term testing in real pianos, the results seems to be promising for practice.

Other editing:

• We removed the duplicate citations (Hill no.4 and Sandberg no. 22).

• We added citation (Zhu et al.) in introduction - page 2.

• Additionally, we added the used duration time of thermal treatment - 8 and 10 hours in page 3.

• Text reorganization was performed. Figure 7 has been moved after the paragraph ending with "Fig.7 shows the results" (page 7 and 8). Text continues with paragraph dealing with suggested discussion of acoustic properties.

Reviewer 2 Report

The article has a logical structure. An overview of the current state is the basis for creating an experiment. The test material is characterized, the research method is explained and add way of evaluation, the results are presented in tables. The research is current.

I think that autors would like to dusscus ther results with another references, becouse this version 3th chapter is only Results.

Increased attention needs to be paid to references. The order of 15 to 20 needs to be adjusted.
EN 13183-1 is not in the references.
The abbreviations HT need to be explained in chapter 2.

Author Response

Dear Reviewers,

Thank you very much for your careful and thorough reading of our manuscript, we really appreciate your positive feedback and would like to thank you for constructive comments and helpful suggestions in the first round of review.

We have proofread the manuscript again and we have tried to do our best to respond to the points raised. As indicated below, we have checked all the specific comments and made the necessary changes accordingly to your indications.

Thank you again for your time.

Jan Tippner, Mendel university in Brno, 30th Decemeber 2020

Reviewer II:

• We wrote the discussion, in order to compare measuring data with other authors. It was added discussion in the title "hygroscopicity" (page 5); "swelling" (page 6); "acoustical properties" (page 6) and "weight loss" (page 9).

• We made the adjusting of no. 15 to 20 in references: no. 16 and 17 - several errors were found in the form of missing spaces, sorry for this. In next, References were now totally reorganized due to improving of the discussion and we used all available information according to the MDPA citation standards. We believe there are now no other errors in the citations.

• Citations of Czech/European standard EN 13183-1 was added in References. We have added the citation of this standard as a no. [21].

• The abbreviations HT (High Temperature) was explained in chapter 2.

Other editing:

• We removed the duplicate citations (Hill no.4 and Sandberg no. 22).

• We added citation (Zhu et al.) in introduction - page 2.

• Additionally, we added the used duration time of thermal treatment - 8 and 10 hours in page 3.

• Text reorganization was performed. Figure 7 has been moved after the paragraph ending with "Fig.7 shows the results" (page 7 and 8). Text continues with paragraph dealing with suggested discussion of acoustic properties.

Round 2

Reviewer 2 Report

Authors accepted my comments.